# Disorganized functional architecture of amygdala subregional networks in obsessive-compulsive disorder

Lingxiao Cao [1,2,5], Hailong Li[1,2,5], Jing Liu[1,2], Jiaxin Jiang[3], Bin Li[3], Xue Li[4], Suming Zhang[1,2], Yingxue Gao[1,2], Kaili Liang[1,2], Xinyue Hu[1,2], Weijie Bao[1,2], Hui Qiu[1,2], Lu Lu[1,2], Lianqing Zhang[1,2], Xinyu Hu[1,2], Qiyong Gong [1,2] & Xiaoqi Huang [1,2✉]

A precise understanding of amygdala-centered subtle networks may help refine neuro-circuitry models of obsessive-compulsive disorder (OCD). We applied connectivity-based parcellation methodology to segment the amygdala based on resting-state fMRI data of 92 medication-free OCD patients without comorbidity and 90 matched healthy controls (HC). The amygdala was parcellated into two subregions corresponding to basolateral amygdala (BLA) and centromedial amygdala (CMA). Amygdala subregional functional connectivity (FC) maps were generated and group differences were evaluated with diagnosis-by-subregion flexible factorial ANOVA. We found significant diagnosis × subregion FC inter-actions in insula, supplementary motor area (SMA), midcingulate cortex (MCC), superior temporal gyrus (STG) and postcentral gyrus (PCG). In HC, the BLA demonstrated stronger connectivity with above regions compared to CMA, whereas in OCD, the connectivity pattern reversed to stronger CMA connectivity comparing to BLA. Relative to HC, OCD patients exhibited hypoconnectivity between left BLA and left insula, and hyperconnectivity between right CMA and SMA, MCC, insula, STG, and PCG. Moreover, OCD patients showed reduced volume of left BLA and right CMA compared to HC. Our findings characterized disorganized functional architecture of amygdala subregional networks in accordance with structural defects, providing direct evidence regarding the specific role of amygdala subregions in the neurocircuitry models of OCD.

[1] Huaxi MR Research Center (HMRRC), Functional and Molecular Imaging Key Laboratory of Sichuan Province, Department of Radiology, West China Hospital of Sichuan University, 610041 Chengdu, China. [2] Psychoradiology Research Unit of the Chinese Academy of Medical Sciences (2018RU011), West China Hospital of Sichuan University, Chengdu, Sichuan, China. [3] Mental Health Center, West China Hospital of Sichuan University, 610041 Chengdu, China. [4] College of Physical Science and Technology, Sichuan University, Chengdu, Sichuan, China. [5]These authors contributed equally: Lingxiao Cao and Hailong Li. ✉email: julianahuang@163.com

As the key structure of emotional circuit, the amygdala has been incorporated into the recent neurocircuitry model of obsessive-compulsive disorder (OCD) for its essential role in mediating fear and anxiety and its rich interactions with the classic cortico-striatal-thalamo-cortical circuit[1–3]. Moreover, amygdala-centered network dysfunction has been recognized as an updated model underlie the pathophysiology and symptomatology of OCD[4]. Within this model, the amygdala is hyper-responsive to fear and uncertainty and lacks optimal functional interactions with prefrontal regions, leading to production and/or maintenance of fear in the context of OCD triggers.

However, few studies have examined the functional integrity of major amygdala networks in relation to OCD using seed-based functional connectivity (SBFC) analysis, which is a popular tool to characterize the functional architecture of spontaneously coupled brain networks in psychiatric disorders[5]. To the best of our knowledge, only three studies have investigated the alterations of amygdala functional connectivity in OCD patients compared with healthy controls (HC) using whole-brain SBFC approach with two studies failing to find any significant group differences between OCD patients and healthy control subjects[6,7]. The only one demonstrated decreased intrinsic connectivity between the right amygdala and the right postcentral gyrus in medicated OCD patients[8]. Nevertheless, all these studies have examined the amygdala as a single, homogeneous region, disregarding the separable functions and connectivity profiles of its distinct subregions.

The amygdala is composed of structurally and functionally discrete subnuclei, commonly grouped into the basolateral amygdala (BLA) and centromedial amygdala (CMA) complexes[2,9,10]. Each of them form dissociable brain networks through their specialized patterns of integration with other cortical and subcortical regions to support distinct amygdala functions[11,12]. Accumulated evidence has emerged to imply the specific role of amygdala subregional functional networks in OCD. For example, an animal study pointed to the unique BLA to the medial prefrontal cortex circuit that controls OCD-like checking symptoms[13]. Moreover, different treatment modalities of OCD may target specific amygdala subregional functional networks. For instance, cognitive behavioral therapy (CBT) is the first line treatment option for OCD patients[14], and previous studies have consistently demonstrated that functional connectivity between the BLA and ventromedial prefrontal cortex could predict CBT outcomes in OCD patients[15,16]. In addition, it has been reported that deep brain stimulation (DBS), as an emerging treatment to refractory OCD, might attenuate mood and anxiety symptoms in OCD by modulating functional networks involving BLA and insula[17]. Therefore, clarifying alterations of amygdala subregional functional networks may advance our understanding of neurobiological mechanisms underlying OCD.

Furthermore, anxiety-related cerebral abnormalities often involve coinciding alterations in functional connectivity and structural properties of amygdala subregions[11,18,19]. A coupling between brain structure and function might be expected as neural network interaction and information processing depend heavily on structural features of neurons (e.g., size, configuration, and arrangement)[20]. As such, a conjoint examination of amygdala functional connectivity and structure seems necessary for a deeper understanding of OCD pathophysiology.

Thus, in the current study, we recruited a relatively large sample of medication-free OCD patients without comorbidity, aiming to characterize functional architecture of amygdala-centered subtle networks in OCD with connectivity-based parcellation (CBP) derived subregions of the amygdala. In addition, we extracted the volume of amygdala subregions to determine whether functional aberrations were accompanied with structural changes. Given earlier reports of BLA functional connectivity in relation to OCD[13,15–17], we cautiously hypothesized that patients with OCD would bear BLA-specific anomalies in amygdala intrinsic networks with prefrontal cortex and insula. As abnormal functional connectivity and structure of amygdala subregions tend to accompany each other[11,18,19], we also hypothesized that abnormal BLA connectivity would coincide with altered volume of the BLA in OCD. We did not formulate specific hypotheses regarding the other amygdala subregions because of the scarcity of prior reports on the involvement of other amygdala subregions in OCD. We hope this study could elucidate specific changes regarding amygdala subregions in OCD with large sample size and controlling for medication and comorbidity confounding effects.

## Results

**Demographic and clinical characteristics**. Demographic and clinical characteristics of the subjects are displayed in Table 1. There was no significant difference between OCD patients and HC regarding age or sex ($P > .05$). For the 92 patients with OCD, the total Y-BOCS score was $21.39 \pm 5.51$, corresponding to moderate and severe OCD symptoms, with obsessive and compulsive scores of $13.01 \pm 5.09$ and $8.38 \pm 5.33$, respectively. The duration of illness was $7.39 \pm 5.54$ years. The HAMA score was $9.12 \pm 4.67$, and the HAMD score was $9.03 \pm 5.24$.

**Parcellations of the amygdala**. Among the two- to five-cluster solution, we considered that the two-cluster solution of dorsal (superior) and ventral (inferior) subregions fitted the input data best for both the left and right amygdala based on the internal validity metrics (see Fig. 1a and Supplementary Fig. 1). The bipartite dorsal-ventral subregions occupied a roughly consistent location to the cytoarchitectonic mapping of the amygdala[21]. The dorsal cluster (orange) resembled the cytoarchitectonically defined CMA, while the ventral cluster (blue) resembled the BLA. The putative BLA cluster showed stronger connectivity with widely distributed cortical areas, encompassing mainly precuneus, posterior cingulate cortex, prefrontal cortex, superior and middle temporal gyrus, precentral and postcentral gyrus, inferior and middle occipital gyrus, and parahippocampal gyrus, whereas the putative CMA cluster exhibited stronger connectivity with multiple subcortical structures, including the striatum, thalamus, midbrain, and cerebellum (see Fig. 1b). This dissociated func-

| Table 1 Demographic and clinical characteristics. | | | |
|---|---|---|---|
| | **OCD (_n_ = 92)** | **HC (_n_ = 90)** | **Comparison** |
| Age, years | 29.42 (8.67) | 28.34 (10.85) | p = .460 |
| Sex (M/F) | 57/35 | 55/35 | p = .907 |
| Duration, years | 7.39 (5.54) | – | – |
| Age of onset, years | 22.03 (7.08) | – | – |
| Y-BOCS | | | |
| Total score | 21.39 (5.51) | – | – |
| Obsessive score | 13.01 (5.09) | – | – |
| Compulsive score | 8.38 (5.33) | – | – |
| HAMA-14 | 9.12 (4.67) | – | – |
| HAMD-17 | 9.03 (5.24) | – | – |

_F female, HAMA Hamilton Anxiety Rating Scale, HAMD Hamilton Depression Rating Scale, HC healthy controls, M male, OCD obsessive-compulsive disorder, Y-BOCS Yale-Brown Obsessive Compulsive Scale._
_Unless otherwise indicated, data are presented as mean (SD)._

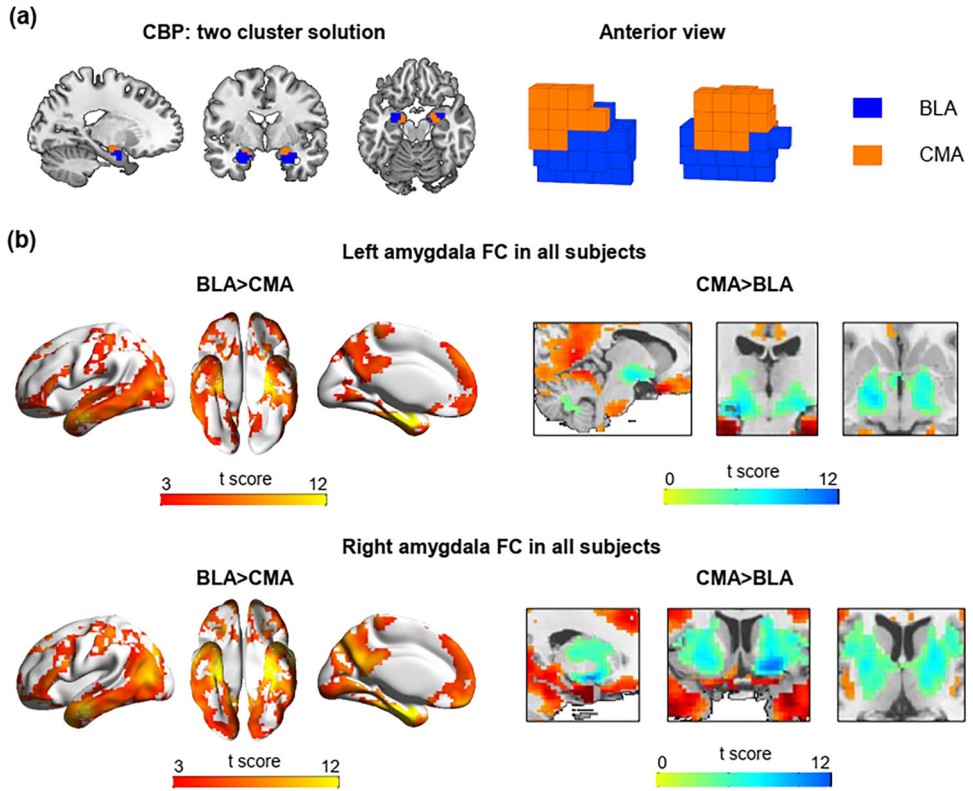

**Fig. 1 Connectivity-based parcellation of the amygdala. a** The two-cluster solution derived by connectivity-based parcellation (CBP). **b** The basolateral cluster (BLA) connectivity was primarily cortical, whereas the centromedial cluster (CMA) connectivity was primarily subcortical. Warm color (red) indicates that BLA connectivity is stronger compared to CMA connectivity whereas cool color (blue) indicates CMA connectivity is stronger compared to BLA connectivity.

tional connectivity pattern is also in line with prior findings of cortical pattern of BLA connectivity and subcortical pattern of CMA connectivity[11,12]. Thus, we considered that CBP-derived amygdala clusters approximately corresponded to the known cytoarchitectonically defined subregions of the amygdala (BLA and CMA). The spatial correlation between the BLA/CMA clusters and their respective parcellation from cytoarchitectonically defined probabilistic maps of the amygdala[22] is presented in Supplementary Fig. 2. Similarity between the individual clustering results and the group-level clustering result on the two-cluster solution are shown in Supplementary Fig. 3.

**Main effect of diagnosis on amygdala functional connectivity**. As shown in Supplementary Fig. 4, relative to HC, OCD patients showed hyperconnectivity between left amygdala and ventromedial prefrontal cortex and middle temporal gyrus, and hypoconnectivity between right amygdala and fusiform.

**Amygdala subregional functional connectivity**. For left amygdala, significant interactions between diagnosis (OCD vs. HC) and subregion (BLA vs. CMA) were demonstrated in bilateral insula. For right amygdala, we found significant diagnosis × subregion interactions in several brain regions including bilateral supplementary motor area (SMA), midcingulate cortex (MCC), superior temporal gyrus (STG), postcentral gyrus (PCG), and right insula. (see Fig. 2a and Supplementary Table 1).

For left amygdala, post hoc analyses revealed that HC demonstrated stronger BLA intrinsic connectivity with bilateral insula compared to CMA, whereas in patients with OCD the CMA showed stronger connectivity with bilateral insula compared to BLA. For right amygdala, stronger BLA connectivity was observed in bilateral SMA, MCC, STG, PCG and right insula compared to CMA in the HC group. While in OCD patients, the CMA exhibited stronger connectivity with the above regions excluding the left PCG. (see Fig. 2a and Supplementary Table 2).

Moreover, there is hypoconnectivity between left BLA and left insula and hyperconnectivity between right CMA and bilateral SMA, MCC, STG, PCG, and right insula in OCD patients compared to HC. (see Fig. 2b and Supplementary Table 3).

**Validation of amygdala subregional functional connectivity**. The results mostly remained consistent whether smooth with a 6 mm FWHM Gaussian kernel or no smooth were applied to the imaging data (see Supplementary Figs. 5, 6). As shown in Supplementary Fig. 7, the same findings attained when using an aggressive head motion control strategy (scrubbing).

The results of exploratory analyses using standard BLA/CMA parcellation[22,23] are shown in Supplementary Fig. 8, which is quite similar to that using CBP-derived BLA/CMA.

**Amygdala structure**. The volumes of whole amygdala were significantly reduced bilaterally in OCD patients relative to HC (left: $p = .037$, $\eta^2 = .024$; right: $p = .012$, $\eta^2 = .035$). We found significantly reduced volume of the BLA (left accessory basal nucleus: $p_{\text{FDR-corrected}} = .018$, $\eta^2 = .042$; right accessory basal nucleus: $p_{\text{FDR-corrected}} < .001$, $\eta^2 = .104$), the CMA (right central nucleus: $p_{\text{FDR-corrected}} = .001$, $\eta^2 = .073$; right medial nucleus: $p_{\text{FDR-corrected}} < .001$, $\eta^2 = .091$), and the cortical nucleus (left: $p_{\text{FDR-corrected}} = .002$, $\eta^2 = .067$; right: $p_{\text{FDR-corrected}} < .001$, $\eta^2 = .18$) in OCD patients compared to HC (see Fig. 3b and Supplementary Table 4).

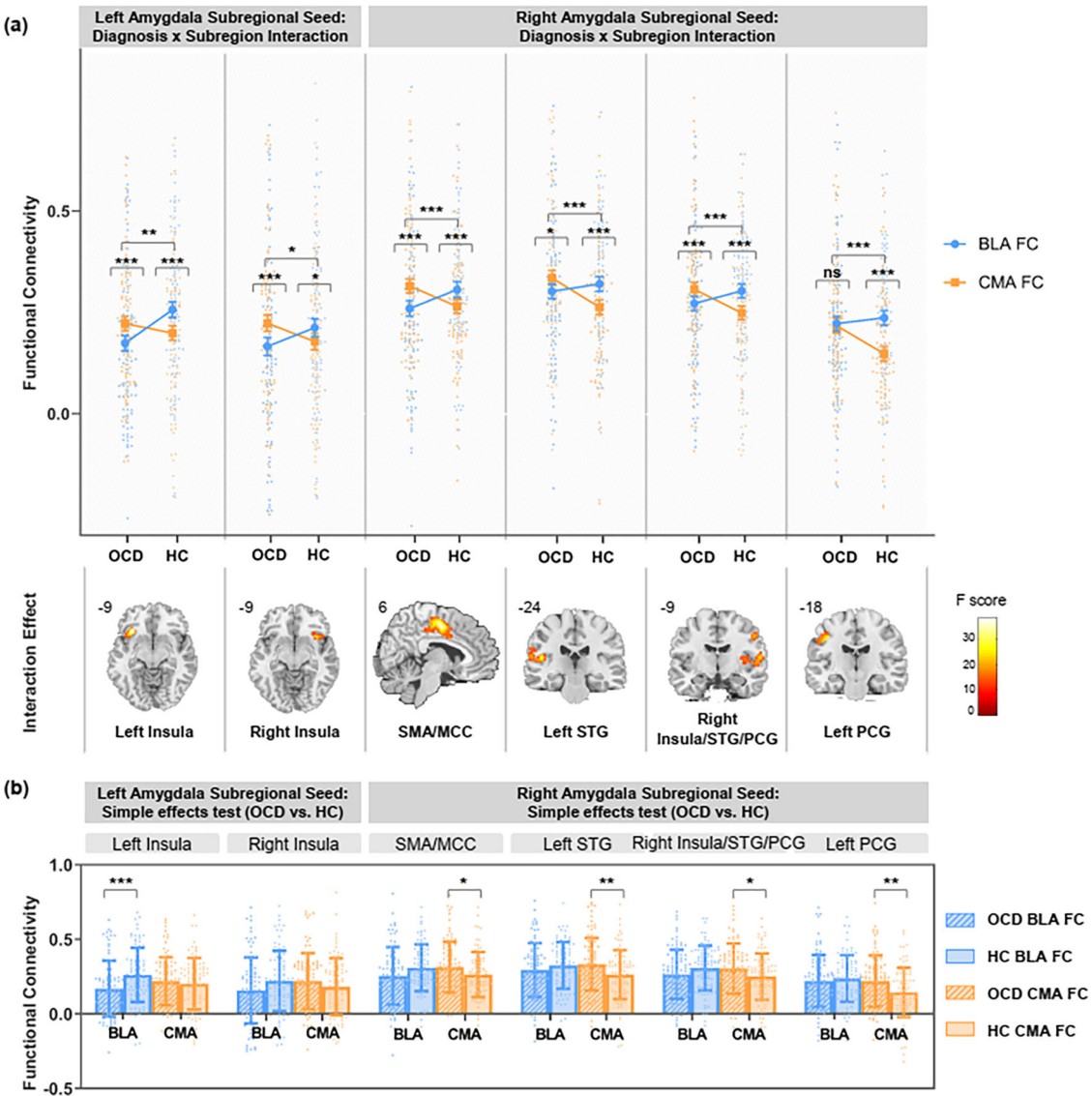

**Fig. 2 Amygdala subregional functional connectivity in OCD and HC. a** Significant interactions between diagnosis (OCD vs. HC) and subregion (BLA vs. CMA) computed separately in left and right amygdala. Error bars represent standard errors of the means. **b** Simple effects tests comparing functional connectivity (FC) between OCD versus HC in each subregion. Significance is indicated for uncorrected *$p < .05$, **$p < .01$, ***$p < .005$. Error bars represent standard deviation. $n = 182$ biologically independent samples were used to derive statistics. MCC midcingulate cortex, PCG postcentral gyrus, SMA supplementary motor area, STG superior temporal gyrus.

We did not identify any significant correlations between amygdala subregional alterations and clinical characteristics in OCD patients after controlling for multiple comparisons.

## Discussion

To the best of our knowledge, this is the first study to conjointly examine the functional connectivity and structural alterations of amygdala subregions in OCD. In HC group, we found stronger BLA connectivity with several cortical regions including insula, SMA, MCC, STG and PCG compared to CMA, whereas in OCD, the connectivity pattern reversed to stronger CMA connectivity comparing to BLA. The disruptions in differentiated BLA-CMA functional connectivity pattern in OCD patients were driven by hypoconnectivity between left BLA and left insula, and hyper-connectivity between right CMA and bilateral SMA, MCC, STG, PCG, and right insula. Interestingly, these connectivity changes were accompanied with significant volume reductions in

amygdala subnuclei in OCD patients. Taken together, our work provides a comprehensive profile of amygdala subregions involvement in patients with OCD from both functional and structural aspect which advanced the neurocircuitry model of OCD.

Prior studies investigating the amygdala subregional functional connectivity usually relied on cytoarchitecture-based templates of the amygdala[11,12,24], which were defined using limited samples[21]. Such anatomically defined partitions might not conform well to the functional boundaries in different populations and thus violate the functional network estimation[25,26]. In the current study, we adopted a CBP technique to conceptualize amygdala functional subregions, which may achieve better performance in terms of resting-state signal homogeneity and provide a good representation of our specific sample for subsequent functional connectivity analyses[27]. More importantly, the inconsistency in average subject age, MRI scanner, and MRI preprocessing strategy across studies makes group-specific parcellation being

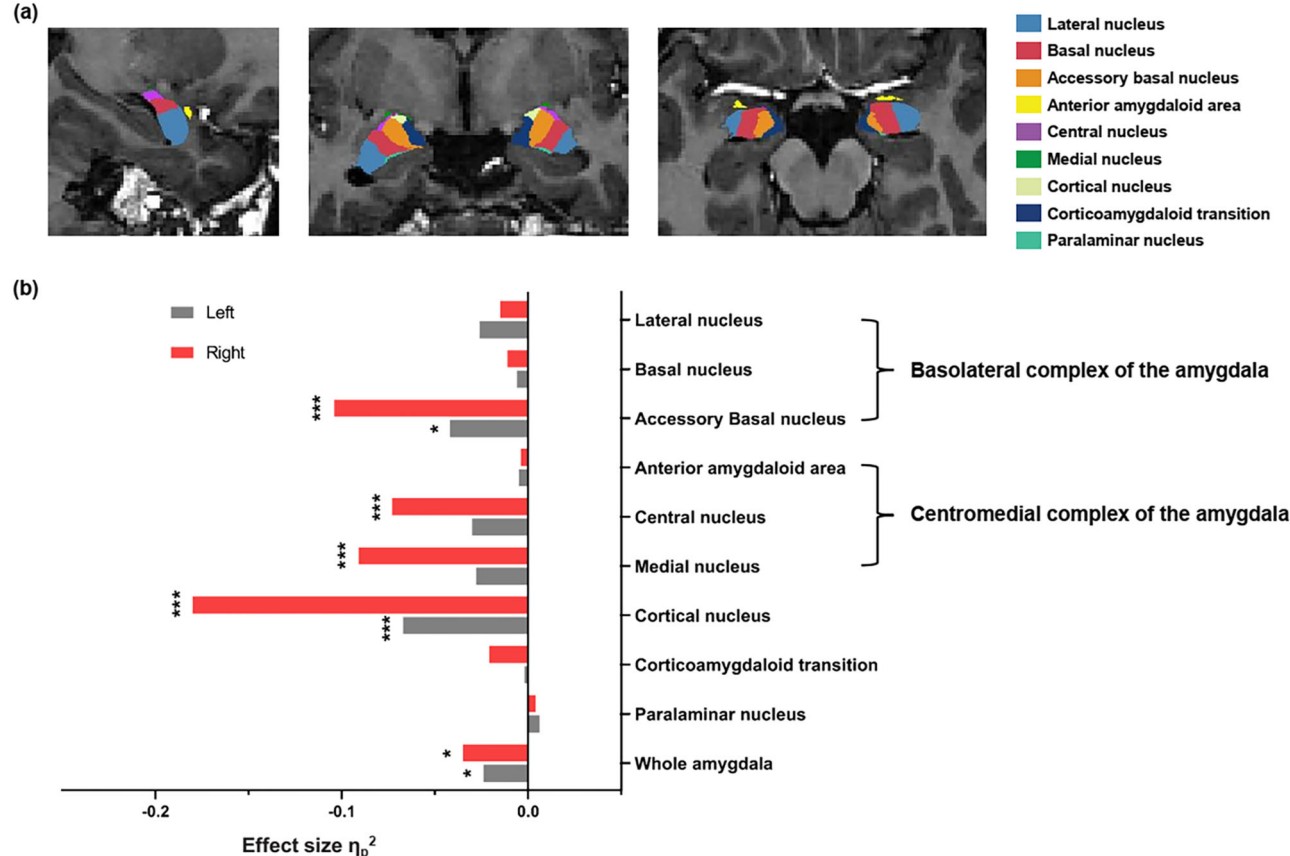

**Fig. 3 Amygdala subregional volume in OCD and HC. a** An example of amygdala segmentation in a healthy control subject. **b** Effect size for the left (gray) and right (red) amygdala nuclei. Positive effect size indicates larger volume in obsessive-compulsive disorder (OCD) group compared to healthy control (HC) group, and negative effect size indicates smaller volume in OCD group compared to HC group. Significance is indicated for false discovery rate–corrected *$p < .05$; **$p < .01$; ***$p < .005$. $n = 182$ biologically independent samples were used to derive statistics.

superior to the standard template in practicing precision medicine in psychiatry.

Using this connectivity-based framework with clustering algorithms, we found that bipartite subregions of the amygdala fitted the input data best and each subregion exhibited preferential functional connectivity with specialized target brain regions. Specifically, one amygdala subregion that is located ventral and associated with exclusively cortical connectivity resembles BLA; while the other subregion that occupies dorsal part and connects almost exclusively with subcortical regions resembles CMA.

It's not surprising that the amygdala subregions we got parallels cytoarchitectonically defined amygdala partitions[21] regarding spatial extent and functional connectivity patterns, since neural network interaction depends heavily on structural properties of neuronal cells (e.g., size, configuration, and arrangement)[20] and that CBP yield clusters cover known anatomical parcellations also verified that cytoarchitecture is the basic structure of brain organization related to intrinsic functional connectivity[28].

Comparing to HC, patients with OCD demonstrated disorganized patterns of differentiated BLA-CMA functional connectivity, specifically with several brain regions including insula, SMA, MCC, STG, and PCG. Both animal models of amygdala circuits[2,9] and previous fMRI studies of amygdala connectivity[11,12] supported the specialized connectivity profiles of the amygdala subregions which involve distinct functions. Accordant with its critical role in associative learning processes, the BLA connects with widely distributed cortical regions including temporal regions, precentral gyrus, postcentral gyrus,

cingulate gyrus, and insula, facilitating their involvement in detection, integration, and regulation of emotionally salient stimuli[1,29]. In contrast, the CMA is the primary output site to orchestrate behavioral and physiological aspects of emotion processing through interactions with subcortical structures. We found increased functional connectivity between the CMA and several cortical regions in OCD patients compared with HC, including insula, SMA, MCC, STG, and PCG. We guess such alterations might be the cause of the reversed pattern of amygdala subregional functional connectivity in OCD patients (i.e., in HC, the BLA demonstrated stronger connectivity with the above cortical regions compared to CMA, whereas in OCD, the connectivity pattern reversed to stronger CMA connectivity comparing to BLA). The disturbance of subregional specialized functional connectivity patterns of the amygdala with these specific regions in OCD patients may underlie the impairments in myriad functions, including integration of sensory information, generation and control of motor behaviors and behavioral flexibility[4].

We found altered amygdala subregional connectivity with the insula in OCD patients. The insula plays imperative roles in the perception of internal feelings and integrating such information with salient environmental inputs via reciprocal connections with the amygdala[30]. In OCD, the insula has been primarily related to disgust sensitivity[31], while animal studies suggest that it also involved in maladaptive impulse control that may facilitate the development of compulsive behaviors[32]. A recent meta-analysis reported hyperactivity of the amygdala during emotional processing in OCD, and that comorbidity with mood and anxiety

disorders was associated with even higher activation in the right amygdala and insula[33].

Importantly, our results illuminate subregion-specific alterations regarding amygdala networks with the insula (i.e., hypoconnectivity between left BLA and left insula, and hyperconnectivity between right CMA and right insula) in OCD patients. The CMA is a prominent downstream structure of the insula, and the insula-CMA circuit might be responsible for the establishment of appropriate behavioral response[34]. Enhanced connectivity between the CMA and insula reported in the current study may thus reflect excessive and persistent behavior that characterize OCD. Using ten patients with treatment-refractory OCD in DBS on and DBS off states and eleven controls, Fridgeirsson and colleagues found that BLA-insula connectivity tended to be higher in OCD patients than controls when DBS was switched off[17]. They also found that DBS-related increase in BLA-insula connectivity was associated with increase in mood and anxiety symptoms. However, our results revealed decreased functional connectivity between the BLA and insula in OCD patients. We postulated that this discrepancy may be attributed to the differences in sample characteristics. Refractory OCD patients in Fridgeirsson's study may not be representative of the larger population of patients with OCD, and the small sample size was fragile to be influenced of concurrent medication usage, which had been proved to affect neuroimaging findings in OCD[35]. While the present study used a large sample and controlled for confounding variables, such as medication and comorbidity effects, so the results are more likely to present the characteristic of amygdala subregional FC in OCD.

OCD patients showed increased CMA connectivity with the STG, which is critical in the perception of emotions in facial stimuli[36]. The STG exhibits interactive pathways with the amygdala that form regulatory systems involved in social cognition. Based on the evidence of the distinct functions of the amygdala subregions in emotion processing, the CMA was preferentially sensitive to negative visual stimuli[37]. Exaggerated CMA connectivity with STG, as reported here, could speculatively allude to deficits in the recognition of social and emotional cues, specifically fearful facial expressions. This fits well with the OCD literature reporting patients characterized by harm/checking symptoms were more sensitive in facial expression recognition[38]. In sum, our finding may thus suggest a possible mechanism for altered emotion processing in OCD patients, which may underlie the clinical manifestations of OCD.

Relative to HC, OCD patients additionally exhibited hyperconnectivity of right CMA with SMA and postcentral gyrus. The SMA is known to be responsible for linking sensory information to appropriate behavior selection and cognitive control[39]. The postcentral gyrus comprises the primary somatosensory cortex and functions together with the amygdala to integrate somatosensory information with emotional input and link perception of emotional stimuli to actions[40,41]. A diffusion tensor imaging study demonstrated that the reconstructed tracts from the amygdala to motor-related areas (SMA and postcentral gyrus) mainly arise from the basolateral subregion[42]. Exaggerated CMA functional connectivity with SMA and postcentral gyrus in OCD patients as reported here may reflect deficiency in sensory-motor integration processes and control of motor behaviors, which finally manifested as compulsions. Previous research has demonstrated reduced functional connectivity between the right amygdala and the right postcentral gyrus in OCD patients[8]. Here, our results specify the association with the CMA subregion which may play a vital role in amygdala-postcentral gyrus connectivity dysfunction in OCD.

Our results also revealed that OCD patients had hyperconnectivity between right CMA and MCC, more specifically, the posterior division (pMCC). Disruption of anterior division of MCC (aMCC) in OCD has been well-established for its unique role in decision making[43]. Our results otherwise suggest the importance of the neglected pMCC in relation to OCD. The pMCC plays a crucial role in multisensory action monitoring and motor responses, including response to stimuli that are effective in alerting to a mismatch between expected and actual motor outcomes[43]. The CMA subregion serves as the amygdala's output site to orchestrate behavioral response of emotion processing. We cautiously assumed that enhanced functional connectivity between the CMA and pMCC may be implicated in excessive stimulus-response-based behaviors in OCD.

In conjunction with amygdala subregional functional network dysfunction, structural analyses demonstrated reduced volume of the bilateral BLA and right CMA subnuclei in OCD patients. This may contribute to perturbed behavioral flexibility and fear conditioning in OCD[44] and support the amygdala-centered model regarding the pathophysiology of OCD. Of interest to the present study, abnormal functional connectivity of left BLA and right CMA in OCD was accompanied with the emergence of volume reductions. Since network communication depends heavily on the structural properties of neuronal cells (e.g., size, configuration, and arrangement), we tentatively hypothesized that amygdala subregional network dysfunction in OCD is related to structural defects, though the exact mechanism about how the change in structure impact human brain function is still poorly understood. Future advances in imaging technology, and greater dialogue between human neuroimaging studies and cellular/molecular neuroscience, could further our understanding of the complicated structure-function relationship.

We didn't find any significant correlations between amygdala subregional alterations and clinical characteristics in OCD patients. Although this could relate to not enough variation in clinical variables, it may also be suggestive of a complex interplay between amygdala alterations and clinical characteristics of OCD involving other moderating and mediating factors. Future work is needed to further our understanding of this complex relationship and elucidate the pathology of OCD. Furthermore, we cautiously hypothesized that amygdala alterations we documented might be more representative of OCD vulnerability and diagnosis rather than its duration or symptom severity.

Several limitations in this study need to be considered while suggesting future directions. First, the exclusion of comorbidity in our sample would help identify OCD-specific features but reduce the generalizability of our results. Future studies with more varied OCD population may be needed to verify the results. Second, the structural parcellation used don't match exactly the functional one and the results should be interpreted with caution. Third, although the amygdala subregions conceptualized from CBP technique quite resemble the cytoarchitectonically defined BLA and CMA, they were not overlap exactly, which makes interpretation of the findings dubious. Fourth, using group-specific parcellation might reduce the generalizability of our findings but outperforms the standard template in terms of providing a better representation of specific sample for subsequent functional connectivity analysis, especially when there are large inconsistency in average subject age, MRI scanner, and MRI preprocessing strategy across studies. Future studies are in need to train a machine learning algorithm on a very large, high-quality, high-resolution independent dataset, and then use this algorithm to predict (define) amygdala parcellations in individual subjects from a new sample, which will bring us closer to prediction medicine.

This study used a data-driven approach to parcellate the amygdala into two distinct subregions comprising BLA and CMA based on their functional connectivity properties. Relative to HC, OCD patients showed disruptions in amygdala subregional

functional networks with several brain regions including insula, SMA, MCC, STG, and PCG. Meanwhile, reduced volumes of the amygdala subnuclei were found in OCD patients in accordance with FC alteration. To the best of our knowledge, this study for the first time demonstrated the emergence of functional abnormalities accompanied with structural defects of amygdala subregions in OCD, which provides important information regarding the amygdala subregions involvement in the neuro-circuitry model of OCD.

## Methods

**Participants**. A total of ninety-three medication-free OCD patients without comorbidity and ninety-three age- and sex-matched HC were enrolled in the present study. All participants were right-handed (assessed by the Edinburgh Handedness Inventory) and native Chinese speakers. OCD patients were recruited from the Mental Health Center, West China Hospital of Sichuan University with diagnosis determined by consensus between two experienced psychiatrists (B. Li and Y. Yang) using the Structured Clinical Interview for DSM-IV Axis I Disorders (SCID). The exclusion criteria were as follows: (1) age younger than 18 or older than 60 years; (2) psychiatric comorbidity assessed using the SCID; (3) pharmacotherapy or psychotherapy within one month of the image acquisition; (4) current or any history of major physical illness, cardiovascular disease, neurological disorder, or substance abuse or dependence; and (5) pregnancy.

The Yale-Brown Obsessive Compulsive Scale (Y-BOCS) was used to assess the severity of OCD symptoms, whereas the 14-item Hamilton Anxiety Rating Scale (HAMA) and 17-item Hamilton Depression Rating Scale (HAMD) were used to evaluate anxiety and depressive symptoms, respectively. Among the OCD patients, fourteen previously had received medication for OCD (four on clomipramine, three on paroxetine, three on fluoxetine, three on sertraline and one on multiple drugs including clomipramine, paroxetine and quetiapine), but had experienced a washout period of at least four weeks before the image acquisition, while the remaining patients were medication-naïve. OCD patients had not undergone psychotherapy in the form of systematic CBT for OCD prior to MRI data acquisition.

HC was recruited from the local area through advertising posters and screened with the SCID (non-patient edition) to confirm the absence of Axis I psychiatric disorders. HC also reported that their first-degree relatives had no known history of psychiatric disorders. The Research Ethics Committee of West China Hospital, Sichuan University approved the current study, and written informed consent was obtained from all participants after receiving a complete description of the study.

**Image acquisition**. Imaging data were acquired on a 3-T GE Signa EXCITE magnetic resonance imaging (MRI) scanner equipped with an 8-channel phase-array head coil. Prior to scanning, participants were instructed to stay awake with their eyes closed and hold still. Foam pads were used to limit head motion, and earplugs were used to attenuate scanner noise.

A total of 200 gradient-echo echo-planar imaging volumes were acquired with the following parameters: repetition time (TR) = 2000 ms, echo time (TE) = 30 ms, flip angle = 90°, 30 axial slices with an in-plane voxel resolution of $3.75 \times 3.75$ mm$^2$, 5 mm slice thickness with no slice gap, and filed of view (FOV) = $240 \times 240$ mm$^2$.

For anatomical reference and structural analysis, a high-resolution T1-weighted three-dimensional spoiled gradient recall sequence was acquired using the following parameters: TR = 8.5 ms, TE = 3.4 ms, flip angle = 12°, 156 contiguous sagittal slices of 1.0 mm thickness, and FOV = $240 \times 240$ mm$^2$ with an acquisition matrix of $256 \times 256$, which yielded an actual voxel size of $0.93 \times 0.93 \times 1$ mm$^3$.

**fMRI image preprocessing**. The resting-state fMRI data preprocessing was performed using the DPABI software[45]. First, the initial ten volumes were discarded to ensure signal stabilization, and slice timing correction was conducted. Then, the time series of images were realigned using a six-parameter (rigid body) linear transformation. After realignment, individual T1-weighted images were co-registered to the mean functional image and then segmented into gray matter, white matter and cerebrospinal fluid. To minimize the head-motion artifacts, we utilized the Friston 24-parameter model to regress out head motion confounding effects[46,47]. Furthermore, several sources of nuisance signals (white matter signal and cerebrospinal fluid signal) were regressed out to reduce the effects of non-neuronal blood oxygen level-dependent fluctuations. After these corrections, the images were spatially normalized to the standard Montreal Neurological Institute (MNI) template (voxel size = $3 \times 3 \times 3$ mm$^3$). We applied spatial smoothing with a 4 mm full width at half maximum (FWHM) Gaussian kernel to the imaging data in primary analysis to minimize the effects of smoothing as a potential source of contamination of amygdala signals from nearby subcortical structures (i.e., the hippocampus) and also included analyses with smooth (6 mm FWHM Gaussian kernel) and without smooth for validation. Finally, linear trend removal was conducted, and then temporal bandpass filtering (0.01–0.08 Hz) was performed to

account for low-frequency machine magnetic field drifts and high-frequency respiratory and cardiac noise.

To minimize the effects of head motion, we adopted a relatively stringent criteria of (1) < 1.5 mm of spatial movement (2) < 1.5 degree of rotation in any direction, and (3) mean framewise displacement (FD) < 0.2 mm[48]. According to this threshold, participants (OCD patients: 1 out of 93; HC: 3 out of 93) were excluded in the following analysis. Additionally, mean FD was used to address the residual effects of head motion as a covariate in group comparisons. In validation analysis, scrubbing (removing time points with FD > 0.2 mm) was utilized to verify results when using an aggressive head motion control strategy. Consistent with previous literature[49,50], we excluded participants (one OCD patient) with <4 min of scrubbed data.

**Connectivity-based parcellation**. CBPtools[51] was used to segment the left and right amygdala separately into distinct subregions based on the resting-state functional connectivity patterns with the whole-brain gray matter. Here, we conducted amygdala connectivity-based parcellation by using all subjects, aiming to produce group-level consensus parcellation and thus facilitate between-group comparisons. A flow chart of parcellation procedure is presented in Fig. 4.

Briefly, functional connectivity between each amygdala voxel (seed mask, extracted from the FSL distributed Harvard-Oxford maximum likelihood subcortical atlas using a probability threshold of 50%) and every voxel of the whole-brain gray matter (target mask, FSL distributed average MNI 152 T1 whole-brain gray matter group template) was computed to obtain a voxel-wise seed-to-target connectivity matrix for each subject. Connectivity between every pair of voxels within the seed tends to be high and may dominate the clustering, we therefore chose to remove all seed voxels from the target mask to account for this possible confounding effect. The target mask was subsampled to improve computational efficiency.

Then, a k-means clustering (with k from 2 to 5, the k-means++ initialization method, 256 initializations, and a maximum of 10,000 iterations) was applied to each subject's connectivity matrix to assign each amygdala voxel to a cluster, effectively grouping similar voxels based on their connectivity patterns and obtaining individual amygdala parcellation. The k-means algorithm was chosen due to its popularity in CBP literature, and the range of k was chosen after consulting relevant literature on the amygdala parcellation[51,52].

Since the individual-level cluster ids are arbitrary, the individual clustering should be relabeled such that similar clusters get assigned the same cluster ids across subjects. Hierarchical clustering with complete linkage and Hamming distance was applied on individual clustering at each k and Hamming distance was used to take the arbitrary nature of the cluster ids into account. The hierarchical clustering results served as reference for relabeling the individual clustering. The resulting labels were then used to compute the mode for each amygdala voxel, finally serving as the group-level clustering results that best describe all included subjects for each k. Several cluster quality metrics including the Silhouette index, the Davies–Bouldin index, and the Calinski–Harabasz index were obtained to determine the appropriate number of clusters. We also computed the adjusted rand index (ARI) as a similarity measure between individual and group clustering.

**Functional connectivity analysis**. Functional connectivity analyses with the CBP-derived amygdala subregions from each hemisphere as seeds were performed using Resting-State fMRI Data Analysis Toolkit software. For each subject, regional time series within each seed (averaging across all voxels) were extracted from the pre-processing data and then correlated (Pearson's correlation) with the time series of every voxel in the rest of the whole brain to obtain individual-level FC maps. The correlation coefficients were standardized using a Fisher z transformation to obtain z-value FC maps per subject for further statistical analyses.

**Statistics and reproducibility**. Group-level analyses were fulfilled using SPM12. Considering the substantial number of functional neuroimaging studies reporting lateralized amygdala activation[53], voxel-wise diagnosis-by-subregion flexible factorial analyses of variance were conducted in left and right amygdala seed separately, with diagnosis as a between-group factor and subregion as a within-subject factor. Age, sex and mean FD value were entered as covariates to account for their possible confounding effects. To determine whether there is a group difference in the differentiated amygdala subregional FC patterns, interaction effects (diagnosis × subregion) were examined. We also investigated the main effects of diagnosis on amygdala functional connectivity. Significant clusters were estimated using a threshold of $p < .001$ at the voxel level and family-wise-error (FWE) corrections for multiple comparisons with an extent threshold of $p < .025$ (.05/2, amygdala seeds from two hemispheres) at the cluster level. We labeled the resulting area with the Automated anatomical labeling (AAL) atlas 3 as a reference. The FC values were extracted from the clusters showing significant interactions between diagnosis and subregion for further post hoc analysis in SPSS 24.0 using simple effects test.

We replicated the analyses of amygdala subregional functional connectivity with smooth (6 mm FWHM Gaussian kernel), without smooth and with an aggressive head motion control strategy (scrubbing) for validation.

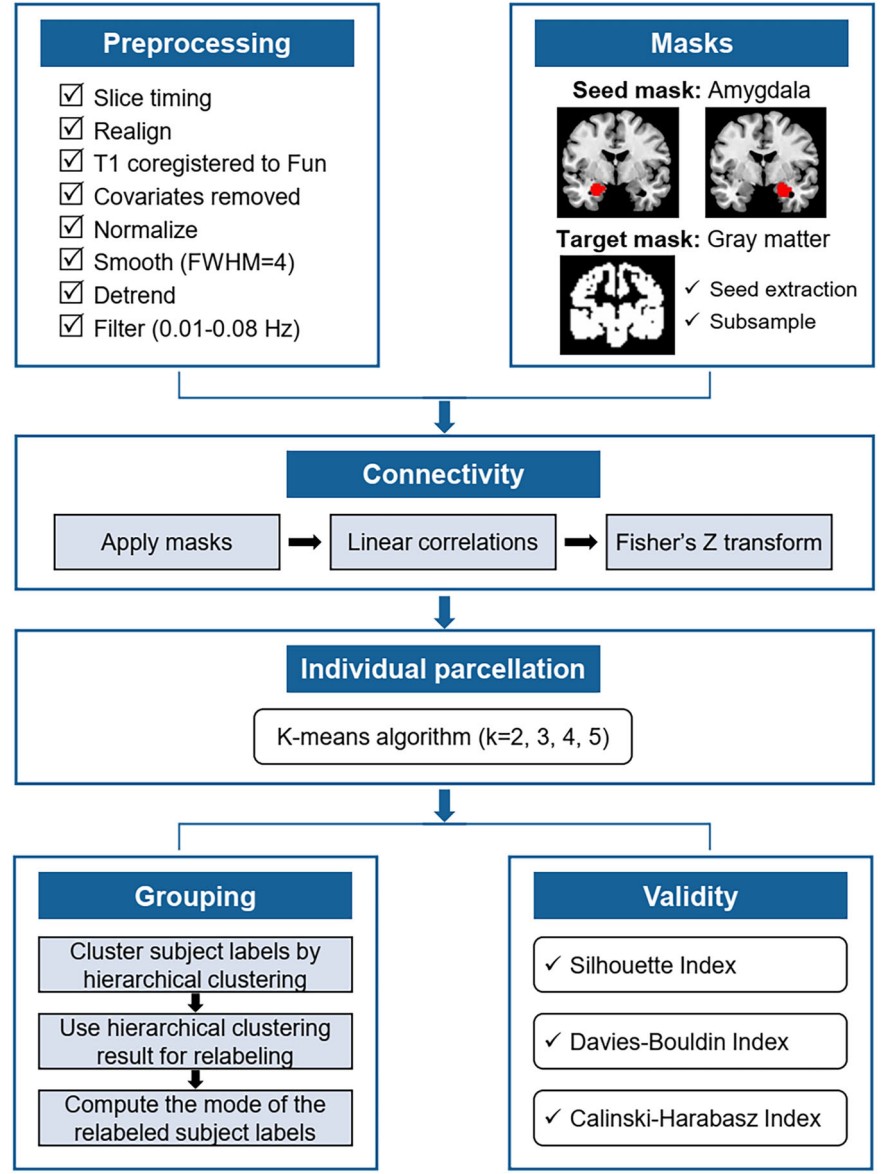

**Fig. 4 Flowchart of the methods.** Procedures of resting-state functional MRI data preprocessing and amygdala parcellation.

**Structural data analysis**. The T1-weighted images were processed using Free-Surfer image analysis suite[54] (version 6.0) with its standard recon-all processing stream. Amygdala segmentation module[55] was used to quantify volumes of nine amygdala subnuclei (the lateral, basal, accessory basal, central, medial, cortical, and paralaminar nuclei as well as the corticoamygdaloid transition area and anterior amygdaloid area) and the whole amygdala in each hemisphere. An example of amygdala segmentation in a healthy control subject is shown in Fig. 3a.

We performed a quality check with visual inspection and statistical outlier detection following ENIGMA quality control protocol. Statistical outlier was determined with < Quartile 1–1.5 times the Interquartile Range (IQR) or > Quartile 3 + 1.5 times the IQR. A participant was removed from the sample if a majority (≥50%) of the subnuclei were statistical outliers. No scans were excluded from the analysis.

We used an analysis of covariance with age, sex, and intracranial volume (ICV) as covariates to test for overall amygdala volume difference between OCD patients and HC. The differences in subnuclei volume between groups were examined using multivariate analysis of covariance with age, sex, and ICV as covariates and FDR correction for multiple subnuclei testing was applied to control type II errors.

**Clinical associations**. To determine whether amygdala subregional functional connectivity alterations were associated with clinical characteristics in OCD patients, partial correlation analyses, controlling for age, sex, and head motion, were used to explore the associations between amygdala subregional FC within regions showing significance and several measures of symptom severity (scores of Y-BOCS, obsessive subscale, compulsive subscale, HAMA, and HAMD), age of onset, and duration of illness. Additional partial correlation analyses with age, sex,

and ICV as covariates were performed to identify clinical associations with volumes of subnuclei that showed significant group differences. To avoid false positive results, we applied false discovery rate (FDR) correction for multiple comparisons.

**Reporting summary**. Further information on research design is available in the Nature Research Reporting Summary linked to this article.

## Data availability
The MRI data that support the findings of this study are available from the corresponding author upon reasonable request. Numerical source data to reproduce all figure panels are available in Supplementary Data 1-2.

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

## Acknowledgements

This work was supported by the grants from 1.3.5 Project for Disciplines of Excellence, West China Hospital, Sichuan University (grant number ZYJC21041), Clinical and Translational Research Fund of Chinese Academy of Medical Sciences (grant number 2021-I2M-C&T-B-097), Natural Science Foundation of Sichuan Province (grant number 2022NSFSC0052), and Sichuan Science and Technology Program (grant number 2021YFS0140). The authors thank Dr. Yanchun Yang from the West China Hospital of Sichuan University for providing valuable help regarding participants recruitment.

## Author contributions

L.C., H.L., Q.G., and Xiaoqi Huang formulated the research questions and designed the study. J.L., J.J., B.L., S.Z., and Xinyu Hu recruited the participants and collected the data. L.C., H.L., X.L., Y.G., K.L., and L.Z. analyzed the data. L.C., H.L., and Xiaoqi Huang wrote the article or revised it. Xinyue Hu, W.B., H.Q., and L.L. reviewed the article. All authors approved the final version of the submitted manuscript.

## Competing interests

The authors declare no competing interests.
