## [Peer Review File · Communications Biology]

Reviewers' comments:

Reviewer #1 (Remarks to the Author):

Thank you for the opportunity to contribute as a reviewer of the paper entitled "Disorganized functional architecture of amygdala subregional networks in obsessive-compulsive disorder". Overall, this is an interesting and well-written paper with several strengths, including a relatively large sample, unmedicated patients with OCD, and rigorous methods. Nevertheless, I have important reservations regarding the parcellation approach employed, in which, I am not an expert. My comments are detailed below. Of note, the lack of page numbering on the manuscript makes referring to different parts of the manuscript difficult. Please include page numbers in future revisions.

Introduction

1. Statement: "To our knowledge, only three studies have investigated the amygdala functional connectivity in patients with OCD using SBFC..."

Comment: Two other studies (refs 15, 16) have also used SBFC (specifically using BLA and CMA seeds), though their approach was seed-to-ROIs rather than seed-to-voxel across the whole brain.

2. Statement: "... an animal study pointed to the unique BLA to the medial prefrontal cortex circuit that controls the checking symptoms of OCD [13]."

Comment: this sentence should be edited to say something like "... that controls OCD-like checking symptoms". This distinction is important because mice don't actually have OCD (animal models may mimic symptoms akin to a disorder, but we cannot say they have a disorder that can only be diagnosed in humans). And since the cited study didn't involve humans with OCD with checking symptoms, we cannot say this circuit controls such symptoms.

3. Statement: "Moreover, the effective treatments of OCD work through targeting specific amygdala subregional functional networks such as deep brain stimulation (DBS)."

Comments:

A) This sentence is confusing the way it is written. Rectify grammar and verb tense concordance. Beyond grammar, the content is also problematic.

B) Presenting DBS as an effective treatment of OCD is a little misleading here. The first line treatments, to which most patients respond, are CBT and pharmacological treatment with SSRIs. DBS is an emerging treatment to which some refractory patients (i.e., who do not respond to other first lines forms of treatment) do respond to some extent. These non-responders may not be representative of the larger population of patients with OCD. In addition, the mechanism of action of this treatment may not be representative of the mechanism of action of other first line treatments to which most patients with OCD respond.

4. Statement: "In addition, it has been reported that functional connectivity between the BLA and ventromedial prefrontal cortex predicted cognitive behavioral therapy outcomes in OCD patients [15,16]."

Comment: Here you have a good opportunity to introduce CBT as the first line treatment and mention the hypothesized underlying mechanism of action of Exposure and response prevention and the role of the amygdala (BLA) - prefrontal regions. Following this, it would be appropriate to expand on treatment options and discuss DBS with the nuances I mentioned in my prior comment, adding further support to the potential importance of the amygdala sub-network FC in the neurobiological

mechanisms of OCD.

5. "We cautiously hypothesized that patients with OCD would bear subregion-specific anomalies in amygdala intrinsic networks with postcentral gyrus, prefrontal cortex, and insula."

Comment: These hypotheses are very vague. Did you predict that both sub-regions would be abnormally connected to these cortical regions? If so, how do you justify such hypothesis when you and cited literature emphasize the distinct networks the different amygdala sub-regions are part of (i.e., BLA being more connected to cortical regions and CMA being more connected to other subcortical regions). Please clearly state your hypotheses with relevant literature that support them. In their current form, these hypotheses seem to have been formulated based on the specific results obtained herein rather than on prior literature.

Methods

6. Participants subsection: Participants were "medication-free OCD patients without comorbidity". What about prior or current treatment in the form of CBT specialized for OCD (i.e., centered on Exposure and Response Prevention)?

7. Amygdala parcellation method:

A) "The resulting labels were then used to compute the mode for each amygdala voxel, finally serving as the group-level clustering results that best describe all included subjects for each k."
-I'm not entirely clear on what this means. Based on this and Figure 2A, it seems like in the end, each subject doesn't have their own individual parcellation of the amygdala, but a group-level parcel (identical for all participants) is used as the final solution for all participants. If my understanding is correct, this method is essentially "template-based SBFC", but instead of using one of the standard templates created based on large unbiased samples with large quantities of high quality (high resolution) data, a group-specific template is created based on individual variations within this specific sample. If this is indeed the case, I do not believe this is a rigorous approach that can yield reproducible and generalizable findings.

At the very least, for the sake of reproducibility, I would like to see validation/exploratory analyses using standard BLA/CMA parcellations (e.g., Pauli et al. A high-resolution probabilistic in vivo atlas of human subcortical brain nuclei. *Sci Data*. 2018; 5: 180063.doi: 10.1038/sdata.2018.63).

8. The structural parcellations used don't match the functional one. Doesn't this cause further interpretation issues? Please discuss.

9. Statement: "...family-wise-error (FWE) corrections for multiple comparisons with an extent threshold of $p < 0.025$ ($0.05/2$, amygdala seeds from two hemispheres) at the cluster level."

Comment: Why examine each hemisphere in two separate analyses? Why not using hemisphere as another within-subject factor in a 2x2x2 ANOVA design (group x amygdala subregion x hemisphere)?

10. Clinical associations: A lot of analyses are planned but there is no mention of correction for multiple tests until the results section. Please mention that correction (and what type) will be applied to these analyses in the Methods section.

Results

11. Statement: "The bipartite dorsal-ventral subregions occupied a roughly consistent location to the cytoarchitectonic mapping of the amygdala [30]. The dorsal cluster (orange) resembled the cytoarchitectonically defined CMA, while the ventral cluster (blue) resembled the BLA."

Comment: I would like to see the exact overlap between the BLA and CMA clusters obtained here and their respective parcellations from a standard atlas (e.g., Pauli et al. A high-resolution probabilistic in vivo atlas of human subcortical brain nuclei. *Sci Data*. 2018; 5: 180063.doi: 10.1038/sdata.2018.63).

12. Statement: "The putative BLA cluster showed stronger connectivity with widely distributed cortical areas, encompassing mainly precuneus, posterior cingulate cortex, prefrontal cortex, superior and middle temporal gyrus, precentral and postcentral gyrus, inferior and middle occipital gyrus, and parahippocampal gyrus, whereas the putative CMA cluster exhibited stronger connectivity with multiple subcortical structures, including the striatum, thalamus, midbrain, and cerebellum"

Comment: Maybe I missed this somewhere in the Methods, but what atlas did you use to label these regions?

13. Simple effects tests comparing functional connectivity between OCD versus HC in each subregion (Figure 3B): 12 post-hoc tests were conducted. With a Bonferroni correction, $p < .004$ would be considered significant. Please only present corrected findings or clearly mention what findings would survive proper correction.

Discussion

14. Statements: "In current study, we adopted a novel CBP technique to conceptualizing amygdala functional subregions, which may achieve better performance in terms of resting-state signal homogeneity and provide a good representation of voxel-wise data for subsequent functional connectivity analyses [34]."[...] "However, the existing individual difference makes group-specific parcellation being superior to the template-based SBFC in practice precision medicine in psychiatry"

Comment: Echoing my prior comments RE this parcellation method, I think this is an interesting approach but potentially problematic at several levels. First, although the amygdala sub-regions identified resembles the BLA and CMA, they may not overlap very accurately. This makes interpretation of the findings dubious.

Second, I would agree that parcellation schemes based on individual-specific FC patterns are superior to approaches using templates based purely on cytoarchitectonic boundaries. Many recent parcellation schemes follow this approach and have the advantage of being created based on large samples (like the publicly available Human Connectome Project dataset), with a large amount of high-quality, high-resolution data. Herein, the parcellation is based on a relatively small sample, including patients with OCD exhibiting smaller amygdala volumes, predominantly male. I believe the group-level parcels thus achieved are likely biased by the specific characteristics of the sample and thus restrict the reproducibility and generalizability of the findings.

An even superior approach would be to train a machine learning algorithm on a very large, high-quality, high-resolution independent dataset, and then use this algorithm to predict (define) amygdala parcellations in individual subjects from a new sample. This is, in my opinion, the approach that will bring us closer to prediction medicine.

15. Statement: "In contrast, the CMA is the primary output site to orchestrate behavioral and physiological aspects of emotion processing through interactions with subcortical structures." [...] "A diffusion tensor imaging study demonstrated that the reconstructed tracts from the amygdala to motor-related areas (SMA and postcentral gyrus) mainly arise from the basolateral subregion [48]"

Comment: How do you reconcile this with all your findings involving connections between the CMA and cortical regions (i.e., insula, STG, SMA and postcentral gyrus)? This pattern of results deepens my doubts about the accuracy/validity of the amygdala parcellation scheme used herein. Please discuss and address these contradictions more clearly.

16. Statement: "A previous study has demonstrated that improvement of mood and anxiety in patients with OCD after DBS treatment was associated with BLA-insula functional connectivity [14]. Thus, our finding of diminished BLA-insula connectivity could potentially be related to the treatment mechanism of OCD."

Comment: The direction of the findings from this cited paper contrasts with that of the findings herein. Specifically, they found that BLA-insula connectivity tended to be HIGHER in patients than controls when DBS was switched off (whereas herein, BLA-insula was DECREASED in OCD). They also found that DBS decreased the impact of the amygdala on the insula and that DBS-related increase in BLA-insula connectivity was associated with increase in mood and anxiety symptoms. Please discuss these discrepancies.

17. Please discuss potential explanations for your lack of BLA-vmPFC findings, consistent with findings from Fridgerisson et al. 2020; Cyr et al. 2020; and Fullana et al. 2017 [refs 14-16].

18. Minor comment: This is a pet-peeve of mine... Because p-value ranges between 0 and 1, the convention is to not put any decimal before the period (e.g., $p < .05$, NOT $p < 0.05$).

Marilyn Cyr (signed review)

Reviewer #2 (Remarks to the Author):

Disorganized functional architecture of amygdala subregional networks in obsessive-compulsive disorder
By Lingxiao Cao, Hailong Li and colleagues (COMMSBIO-22-0675-T)

Summary: This is an interest resting state functional connectivity MRI study that investigated the role of different networks encompassing subregions of amygdala in OCD. In general, authors found that OCD patients compared to controls presented lower connectivity between left BLA and left insula and greater connectivity between right CMA and SMA, MCC, insula, STG and PCG. Reduced gray matter volumes in the left BLA and right CMA were also observed in OCD patients versus controls.

Significance: This study is interesting and well methodologically designed. It covers an under-studied area and addresses a critical issue in an attempt to better characterize neurocircuitry models of OCD having subregions of amygdala as hubs.

Strengths: This paper has a number of strengths:

- 1) As acknowledged, this is the first functional connectivity study that attempted to investigate subregions of the amygdala in OCD, contributing to characterize different brain circuits in this disorder.
- 2) Patients were treatment-free and most patients were treatment-naive.
- 3) The exclusion of patients with comorbidities enhances the internal validity of the findings in relation to OCD itself.
- 4) Authors also investigated brain volume alterations.
- 5) Authors used a novel method to functionally parcellate the amygdala.

Areas needing more detailed coverage:

- 1) Did authors have a prior hypothesis in regards to the direction of the functional connectivity and volume findings?
- 2) What are the advantages of using CBP tools to segment the amygdala in relation to the other methods? Authors should provide more information about this topic.
- 3) Given that the volumes of the amygdala were different between OCD patients and controls, I wonder whether authors included amygdala volume as a covariate to account for its possible confounding effect in the functional connectivity analysis. In addition, were there any associations between amygdala volume and functional connectivity patterns in OCD and controls. These approaches may shed light on the relationship between brain structure and function.
- 4) Authors should provide a discussion about the lack of significant correlations between amygdala subregional alterations and clinical characteristics in OCD patients and explore the implications of such findings in the pathology of OCD.

Responses to Reviewers' Comments

We thank the reviewers for their encouraging, insightful and helpful comments. We feel that responding to these comments has allowed us to considerably clarify and strengthen our paper. The following is a summary of our detailed responses to the reviewers' comments (reviewers' comments are shown in italic). The main changes have been marked in blue in the revised manuscript.

Reviewer #1:

Thank you for the opportunity to contribute as a reviewer of the paper entitled "Disorganized functional architecture of amygdala subregional networks in obsessive-compulsive disorder". Overall, this is an interesting and well-written paper with several strengths, including a relatively large sample, unmedicated patients with OCD, and rigorous methods. Nevertheless, I have important reservations regarding the parcellation approach employed, in which, I am not an expert. My comments are detailed below. Of note, the lack of page numbering on the manuscript makes referring to different parts of the manuscript difficult. Please include page numbers in future revisions.

Response: We appreciate the encouraging and thoughtful comments, which helped us to considerably clarify and strengthen our paper. In this revision, we have added page numbers to the manuscript for ease of referring to different parts of the manuscript.

Introduction

1. Statement: "To our knowledge, only three studies have investigated the amygdala functional connectivity in patients with OCD using SBFC..."

Comment: Two other studies (refs 15, 16) have also used SBFC (specifically using BLA and CMA seeds), though their approach was seed-to-ROIs rather than seed-to-voxel across the whole brain.

Response: We had mentioned these two studies (refs 15, 16) in the next paragraph to illustrate the importance of amygdala subregional FC in OCD. We have revised this sentence more clearly: "To our knowledge, only three studies have investigated **the**

alterations of amygdala functional connectivity in OCD patients compared with HC using whole-brain SBFC approach...”

2. *Statement: “... an animal study pointed to the unique BLA to the medial prefrontal cortex circuit that controls the checking symptoms of OCD [13].”*

Comment: this sentence should be edited to say something like “... that controls OCD-like checking symptoms”. This distinction is important because mice don't actually have OCD (animal models may mimic symptoms akin to a disorder, but we cannot say they have a disorder that can only be diagnosed in humans). And since the cited study didn't involve humans with OCD with checking symptoms, we cannot say this circuit controls such symptoms.

Response: We thank the reviewer for this thoughtful comment. We have edited this sentence as suggested: “... an animal study pointed to the unique BLA to the medial prefrontal cortex circuit that controls **OCD-like checking symptoms** ¹³.”

3. *Statement: “Moreover, the effective treatments of OCD work through targeting specific amygdala subregional functional networks such as deep brain stimulation (DBS).”*

Comments:

A) This sentence is confusing the way it is written. Rectify grammar and verb tense concordance. Beyond grammar, the content is also problematic.

B) Presenting DBS as an effective treatment of OCD is a little misleading here. The first line treatments, to which most patients respond, are CBT and pharmacological treatment with SSRIs. DBS is an emerging treatment to which some refractory patients (i.e., who do not respond to other first lines forms of treatment) do respond to some extent. These non-responders may not be representative of the larger population of patients with OCD. In addition, the mechanism of action of this treatment may not be representative of the mechanism of action of other first line treatments to which most patients with OCD respond.

Response: We apologize for the confusion. This sentence is now clarified as the

following: “Moreover, different treatment modalities of OCD may target specific amygdala subregional functional networks. ...cognitive behavioral therapy (CBT)... In addition, it has been reported that deep brain stimulation (DBS), as an emerging treatment to refractory OCD, might attenuate mood and anxiety symptoms in OCD by modulating functional networks involving BLA and insula ¹⁷.”

4. *Statement: “In addition, it has been reported that functional connectivity between the BLA and ventromedial prefrontal cortex predicted cognitive behavioral therapy outcomes in OCD patients [15,16].”*

Comment: Here you have a good opportunity to introduce CBT as the first line treatment and mention the hypothesized underlying mechanism of action of Exposure and response prevention and the role of the amygdala (BLA) - prefrontal regions. Following this, it would be appropriate to expand on treatment options and discuss DBS with the nuances I mentioned in my prior comment, adding further support to the potential importance of the amygdala sub-network FC in the neurobiological mechanisms of OCD.

Response: We thank the reviewer for this helpful suggestion. We have now added more content to introduce the role of the BLA-prefrontal regions in the hypothesized underlying mechanism of CBT and expand on treatment options to discuss DBS as suggested: “Moreover, different treatment modalities of OCD may target specific amygdala subregional functional networks. For instance, cognitive behavioral therapy (CBT) is the first line treatment option for OCD patients ¹⁴, and previous studies have consistently demonstrated that functional connectivity between the BLA and ventromedial prefrontal cortex could predict CBT outcomes in OCD patients ^{15, 16}. In addition, it has been reported that deep brain stimulation (DBS), as an emerging treatment to refractory OCD, might attenuate mood and anxiety symptoms in OCD by modulating functional networks involving BLA and insula ¹⁷.”

5. *“We cautiously hypothesized that patients with OCD would bear subregion-specific anomalies in amygdala intrinsic networks with postcentral gyrus, prefrontal cortex, and*

insula.”

Comment: These hypotheses are very vague. Did you predict that both sub-regions would be abnormally connected to these cortical regions? If so, how do you justify such hypothesis when you and cited literature emphasize the distinct networks the different amygdala sub-regions are part of (i.e., BLA being more connected to cortical regions and CMA being more connected to other subcortical regions). Please clearly state your hypotheses with relevant literature that support them. In their current form, these hypotheses seem to have been formulated based on the specific results obtained herein rather than on prior literature.

Response: Based on previous studies (refs 13 and 15-17, listed in Introduction, Page 5) on amygdala subregional functional connectivity in OCD, the BLA seems to be more related to OCD, hence we clarified our hypotheses as follows “Given earlier reports of BLA functional connectivity in relation to OCD ^{13, 15-17}, we cautiously hypothesized that patients with OCD would bear BLA-specific anomalies in amygdala intrinsic networks with prefrontal cortex and insula. As abnormal functional connectivity and structure of amygdala subregions tend to accompany each other ^{11,18,19}, we also hypothesized that abnormal BLA connectivity would coincide with altered volume of the BLA in OCD. We did not formulate specific hypotheses regarding the other amygdala subregions because of the scarcity of prior reports on the involvement of other amygdala subregions in OCD. We hope this study could elucidate specific changes regarding amygdala subregions in OCD with large sample size and controlling for medication and comorbidity confounding effects.”

Methods

6. Participants subsection: Participants were “medication-free OCD patients without comorbidity”. What about prior or current treatment in the form of CBT specialized for OCD (i.e., centered on Exposure and Response Prevention)?

Response: OCD patients had not undergone psychotherapy in the form of systematic CBT for OCD prior to MRI data acquisition.

7. *Amygdala parcellation method:*

“The resulting labels were then used to compute the mode for each amygdala voxel, finally serving as the group-level clustering results that best describe all included subjects for each k.”

-I’m not entirely clear on what this means. Based on this and Figure 2A, it seems like in the end, each subject doesn’t have their own individual parcellation of the amygdala, but a group-level parcel (identical for all participants) is used as the final solution for all participants. If my understanding is correct, this method is essentially “template-based SBFC”, but instead of using one of the standard templates created based on large unbiased samples with large quantities of high quality (high resolution) data, a group-specific template is created based on individual variations within this specific sample. If this is indeed the case, I do not believe this is a rigorous approach that can yield reproducible and generalizable findings.

Response: Yes, we used a group-level parcel (identical for all participants) as the final solution, which can complement heterogeneity of subject-level clustering to facilitate between-group comparisons. Thus, this method can be considered as “template-based SBFC” using a group-specific template. We admitted such group-specific template might reduce the generalizability of our findings but outperforms the standard template in terms of providing a better representation of specific sample for subsequent functional connectivity analysis (Eickhoff, Thirion, Varoquaux, & Bzdok, 2015; Eickhoff, Yeo, & Genon, 2018), especially when there are large inconsistency in average subject age, MRI scanner, and MRI preprocessing strategy across studies.

At the very least, for the sake of reproducibility, I would like to see validation/exploratory analyses using standard BLA/CMA parcellations (e.g., Pauli et al. A high-resolution probabilistic in vivo atlas of human subcortical brain nuclei. Sci Data. 2018; 5: 180063.doi: 10.1038/sdata.2018.63).

Response: We appreciate the reviewer’s advice. After consulting previous studies on amygdala subregional functional connectivity (Etkin, Prater, Schatzberg, Menon, & Greicius, 2009; Qin, Young, Supekar, Uddin, & Menon, 2012; Tang et al., 2019), we

found cytoarchitectonically defined probabilistic maps of the amygdala (Eickhoff et al., 2005; Zilles & Amunts, 2010) was the most popular template for creating BLA and CMA region of interest masks, so we chose to use this template with the aim to make comparability to other publications. As is shown in Supplementary Figure 8, the resulting regions from exploratory analyses using standard BLA/CMA parcellations include insula, SMA, MCC, STG and PCG, which is quite similar to that using CBP-derived BLA/CMA.

Supplementary Figure 8 (A) Significant interactions between diagnosis (OCD vs. HC) and subregion (BLA vs. CMA) computed separately in left and right amygdala. (B) Simple effects tests comparing functional connectivity (FC) between OCD versus HC in each subregion. Significance is indicated for uncorrected * $p < .05$; ** $p < .01$; ***, $p < .005$. MCC, midcingulate cortex; PCG, postcentral gyrus; SMA, supplementary motor area; STG, superior temporal gyrus.

8. The structural parcellations used don't match the functional one. Doesn't this cause further interpretation issues? Please discuss.

Response: The structural parcellation was histology-based with border detection techniques, while the functional parcellation used clustering approaches to group voxels based on resting-state functional connectivity. What's more, the image resolution is different between structural MRI and fMRI, so it's not surprising the structural parcellation don't match exactly the functional one.

We found functional connectivity abnormalities in left BLA and right CMA and volumetric reductions in left accessory basal nucleus, right central and medial nucleus in OCD patients. These findings might reveal in part functional connectivity abnormalities coincide with structural defects of amygdala subregions in OCD, though the results should be interpreted with caution. We have added this point as a limitation: "the structural parcellation used don't match exactly the functional one and the results should be interpreted with caution."

9. *Statement: "...family-wise-error (FWE) corrections for multiple comparisons with an extent threshold of $p < 0.025$ ($0.05/2$, amygdala seeds from two hemispheres) at the cluster level."*

Comment: Why examine each hemisphere in two separate analyses? Why not using hemisphere as another within-subject factor in a 2x2x2 ANOVA design (group x amygdala subregion x hemisphere)?

Response: Considering the substantial number of functional neuroimaging studies reporting lateralized amygdala activation (Baas, Aleman, & Kahn, 2004), we performed two separate ANOVA analyses to examine each hemisphere with Bonferroni correction to control false positive results.

10. *Clinical associations: A lot of analyses are planned but there is no mention of correction for multiple tests until the results section. Please mention that correction (and what type) will be applied to these analyses in the Methods section.*

Response: We have added this information in the Methods section: "To avoid false positive results, we applied false discovery rate (FDR) correction for multiple comparisons."

Results

11. Statement: “The bipartite dorsal-ventral subregions occupied a roughly consistent location to the cytoarchitectonic mapping of the amygdala [30]. The dorsal cluster (orange) resembled the cytoarchitectonically defined CMA, while the ventral cluster (blue) resembled the BLA.”

Comment: I would like to see the exact overlap between the BLA and CMA clusters obtained here and their respective parcellations from a standard atlas (e.g., Pauli et al. A high-resolution probabilistic in vivo atlas of human subcortical brain nuclei. *Sci Data*. 2018; 5: 180063.doi: 10.1038/sdata.2018.63).

Response: We calculated the spatial correlation between the BLA/CMA clusters obtained by CBP technique and their respective parcellation from cytoarchitectonically defined probabilistic maps of the amygdala. The results are shown in Supplementary Figure 2. We found that they were quite resembled but not overlap exactly which is understandable. We have addressed this in the Limitation: “Third, although the amygdala subregions conceptualized from CBP technique quite resemble the cytoarchitectonically defined BLA and CMA, they were not overlap exactly, which makes interpretation of the findings dubious.”

Supplementary Figure 2 Spatial correlation between the BLA/CMA clusters obtained by CBP technique and their respective parcellation from cytoarchitectonically defined probabilistic maps of the amygdala.

12. *Statement: “The putative BLA cluster showed stronger connectivity with widely distributed cortical areas, encompassing mainly precuneus, posterior cingulate cortex, prefrontal cortex, superior and middle temporal gyrus, precentral and postcentral gyrus, inferior and middle occipital gyrus, and parahippocampal gyrus, whereas the putative CMA cluster exhibited stronger connectivity with multiple subcortical structures, including the striatum, thalamus, midbrain, and cerebellum”*

Comment: Maybe I missed this somewhere in the Methods, but what atlas did you use to label these regions?

Response: We used AAL atlas 3 to label these regions. This was clarified in the Methods section: “We labeled the resulting area with the Automated anatomical labelling (AAL) atlas 3 as a reference.”

13. *Simple effects tests comparing functional connectivity between OCD versus HC in each subregion (Figure 3B): 12 post-hoc tests were conducted. With a Bonferroni correction, $p < .004$ would be considered significant. Please only present corrected findings or clearly mention what findings would survive proper correction.*

Response: We thought the probability of having 12 significant tests at Bonferroni correction on small sample data might be too rigorous and bear the danger of false negative results. Thus, we applied FDR correction on these p values and the results were presented in Supplementary Table 3. The findings of hypoconnectivity between left BLA and left insula and hyperconnectivity between right CMA and left STG and left PCG in OCD compared with HC remained significant after FDR correction.

Discussion

14. *Statements: “In current study, we adopted a novel CBP technique to conceptualizing amygdala functional subregions, which may achieve better performance in terms of resting-state signal homogeneity and provide a good representation of voxel-wise data for subsequent functional connectivity analyses [34].”*
“However, the existing individual difference makes group-specific parcellation being superior to the template-based SBFC in practice precision medicine in psychiatry”

Comment: Echoing my prior comments RE this parcellation method, I think this is an interesting approach but potentially problematic at several levels. First, although the amygdala sub-regions identified resembles the BLA and CMA, they may not overlap very accurately. This makes interpretation of the findings dubious.

Response: We have calculated the spatial correlation between the CBP-derived amygdala subregions and the cytoarchitecturally defined BLA/CMA and found that they were quite resembled though not overlap exactly. We have added this point as a limitation “Third, although the amygdala subregions conceptualized from CBP technique quite resemble the cytoarchitecturally defined BLA and CMA, they were not overlap exactly, which makes interpretation of the findings dubious.”

Second, I would agree that parcellation schemes based on individual-specific FC patterns are superior to approaches using templates based purely on cytoarchitectonic boundaries. Many recent parcellation schemes follow this approach and have the advantage of being created based on large samples (like the publicly available Human Connectome Project dataset), with a large amount of high-quality, high-resolution data. Herein, the parcellation is based on a relatively small sample, including patients with OCD exhibiting smaller amygdala volumes, predominantly male. I believe the group-level parcels thus achieved are likely biased by the specific characteristics of the sample and thus restrict the reproducibility and generalizability of the findings.

Response: Group-specific parcellation can provide a better representation of specific sample for subsequent functional connectivity analysis than the standard template and thus yield more reliable results. We agreed that using such group-specific parcellation may restrict the generalizability of our findings, so we have added it in the Limitation: “Fourth, using group-specific parcellation might reduce the generalizability of our findings but outperforms the standard template in terms of providing a better representation of specific sample for subsequent functional connectivity analysis, especially when there are large inconsistency in average subject age, MRI scanner, and MRI preprocessing strategy across studies. ”

An even superior approach would be to train a machine learning algorithm on a very large, high-quality, high-resolution independent dataset, and then use this algorithm to predict (define) amygdala parcellations in individual subjects from a new sample. This is, in my opinion, the approach that will bring us closer to prediction medicine.

Response: We totally agreed it and future studies are in need to train a machine learning algorithm on a very large and high-quality independent dataset to predict (define) amygdala parcellations in individual subjects from a new sample, which will bring us closer to prediction medicine. And we have added this issue in the future direction for the manuscript.

15. Statement: “In contrast, the CMA is the primary output site to orchestrate behavioral and physiological aspects of emotion processing through interactions with subcortical structures.” [...]

“A diffusion tensor imaging study demonstrated that the reconstructed tracts from the amygdala to motor-related areas (SMA and postcentral gyrus) mainly arise from the basolateral subregion [48]”

Comment: How do you reconcile this with all your findings involving connections between the CMA and cortical regions (i.e., insula, STG, SMA and postcentral gyrus)? This pattern of results deepens my doubts about the accuracy/validity of the amygdala parcellation scheme used herein. Please discuss and address these contradictions more clearly.

Response: We are sorry for the confusion. Our findings involving connections between the CMA and these cortical regions are the functional connectivity abnormalities presented in OCD patients. Truly, the BLA specifically show stronger connectivity with these cortical regions, whereas the CMA tend to have stronger connectivity with subcortical structures. This dissociated pattern of connectivity between the BLA and CMA identified in this study is in line with numerous previous findings (Etkin et al., 2009; Qin et al., 2012). We found increased functional connectivity between the CMA and these cortical regions in OCD patients compared with HC. We guess such alterations might be the cause of the reversed pattern of amygdala subregional

functional connectivity in OCD patients (i.e., in HC, the BLA demonstrated stronger connectivity with above cortical regions compared to CMA, whereas in OCD, the connectivity pattern reversed to stronger CMA connectivity comparing to BLA). And we have now added some content to clarify this issue.

16. Statement: “A previous study has demonstrated that improvement of mood and anxiety in patients with OCD after DBS treatment was associated with BLA-insula functional connectivity [14]. Thus, our finding of diminished BLA-insula connectivity could potentially be related to the treatment mechanism of OCD.”

Comment: The direction of the findings from this cited paper contrasts with that of the findings herein. Specifically, they found that BLA-insula connectivity tended to be HIGHER in patients than controls when DBS was switched off (whereas herein, BLA-insula was DECREASED in OCD). They also found that DBS decreased the impact of the amygdala on the insula and that DBS-related increase in BLA-insula connectivity was associated with increase in mood and anxiety symptoms. Please discuss these discrepancies.

Response: We have discussed these discrepancies in the Discussion (Page 22) “Using ten patients with treatment-refractory OCD in DBS on and DBS off states and eleven controls, Fridgeirsson and colleagues found that BLA-insula connectivity tended to be higher in OCD patients than controls when DBS was switched off ¹⁷. They also found that DBS-related increase in BLA-insula connectivity was associated with increase in mood and anxiety symptoms. However, our results revealed decreased functional connectivity between the BLA and insula in OCD patients. We postulated that this discrepancy maybe attributed to the differences in sample characteristics. Refractory OCD patients in Fridgeirsson’s study may not be representative of the larger population of patients with OCD, and the small sample size was fragile to be influenced of concurrent medication usage, which had been proved to affect neuroimaging findings in OCD ⁴⁶. While the present study used a large sample and controlled for confounding variables, such as medication and comorbidity effects, so the results is more likely to present the characteristic of amygdala subregional FC in OCD.”

17. Please discuss potential explanations for your lack of BLA-vmPFC findings, consistent with findings from Fridgerisson et al. 2020; Cyr et al. 2020; and Fullana et al. 2017 [refs 14-16].

Response: The inconsistency in the BLA-vmPFC findings could be attributed to the heterogeneity in the sample characteristics (e.g., subject age, medication, and clinical comorbidity) and methods (seed-to-ROIs in previous studies vs. seed-to-voxel across the whole brain in current study). However, we believe the current study with a large sample and controlling for confounding variables bear the advantage of characterizing amygdala subregional FC in OCD not affected by comorbidity and treatments.

18. Minor comment: This is a pet-peeve of mine... Because p-value ranges between 0 and 1, the convention is to not put any decimal before the period (e.g., $p < .05$, NOT $p < 0.05$).

Response: This was corrected in the revised manuscript.

Reviewer #2:

Summary: *This is an interest resting state functional connectivity MRI study that investigated the role of different networks encompassing subregions of amygdala in OCD. In general, authors found that OCD patients compared to controls presented lower connectivity between left BLA and left insula and greater connectivity between right CMA and SMA, MCC, insula, STG and PCG. Reduced gray matter volumes in the left BLA and right CMA were also observed in OCD patients versus controls.*

Significance: *This study is interesting and well methodologically designed. It covers an under-studied area and addresses a critical issue in an attempt to better characterize neurocircuitry models of OCD having subregions of amygdala as hubs.*

Strengths: *This paper has a number of strengths:*

1) As acknowledged, this is the first functional connectivity study that attempted to investigate subregions of the amygdala in OCD, contributing to characterize different brain circuits in this disorder.

2) Patients were treatment-free and most patients were treatment-naive.

3) The exclusion of patients with comorbidities enhances the internal validity of the findings in relation to OCD itself.

4) Authors also investigated brain volume alterations.

5) Authors used a novel method to functionally parcellate the amygdala.

Response: We thank the reviewer for the positive and encouraging comments.

Areas needing more detailed coverage:

1) Did authors have a prior hypothesis in regards to the direction of the functional connectivity and volume findings?

Response: Given the scarcity of prior studies on amygdala in OCD, heterogeneity in the samples used and inconsistent results produced in previous studies, we did not formulate specific hypotheses regarding the direction of the functional connectivity or volume findings. We hope this study could elucidate specific changes regarding amygdala in OCD with large sample size and controlling for medication and comorbidity confounding effects.

2) *What are the advantages of using CBP tools to segment the amygdala in relation to the other methods? Authors should provide more information about this topic.*

Response: We have provided more information regarding the advantage of CBP in Discussion section: “Prior studies investigating the amygdala subregional functional connectivity usually relied on cytoarchitecture-based templates of the amygdala ^{11,12,35}, which were defined using limited samples ³². Such anatomically defined partitions might not conform well to the functional boundaries in different populations and thus violate the functional network estimation ^{36,37}. In current study, we adopted a novel CBP technique to conceptualizing amygdala functional subregions, **which may achieve better performance in terms of resting-state signal homogeneity and provide a good representation of our specific sample for subsequent functional connectivity analyses ³⁸. More importantly, the inconsistency in average subject age, MRI scanner, and MRI preprocessing strategy across studies makes group-specific parcellation being superior to the standard template in practice precision medicine in psychiatry.**” (Page 19)

3) *Given that the volumes of the amygdala were different between OCD patients and controls, I wonder whether authors included amygdala volume as a covariate to account for it possible confounding effect in the functional connectivity analysis. In addition, were there any association between amygdala volume and functional connectivity patterns in OCD and controls. These approaches may shed light in the relationship between brain structure and function.*

Response: The functional parcellation of the amygdala used in the functional connectivity analysis don't match exactly the structural one, so it's challenging to include amygdala subregional volume as a covariate in the functional connectivity analysis and perform correlation analysis between volume and functional connectivity. We consulted other studies on amygdala functional connectivity (Aghajani et al., 2017; Aghajani et al., 2016) and found it was acceptable to not include amygdala volume as a covariate in the functional connectivity analysis.

4) *Authors should provide a discussion about the lack of significant correlations*

between amygdala subregional alterations and clinical characteristics in OCD patients and explore the implications of such findings in the pathology of OCD.

Response: We have provided a discussion accordingly: “We didn’t find any significant correlations between amygdala subregional alterations and clinical characteristics in OCD patients. Although this could relate to not enough variation in clinical variables, it may also be suggestive of a complex interplay between amygdala alterations and clinical characteristics of OCD involving other moderating and mediating factors. Future work is needed to further our understanding of this complex relationship and elucidate the pathology of OCD. Furthermore, we cautiously hypothesized that amygdala alterations we documented might be more representative of OCD vulnerability and diagnosis rather than its duration or symptom severity.”

Reference

- Aghajani, M., Klapwijk, E. T., van der Wee, N. J., Veer, I. M., Rombouts, S., Boon, A. E., . . . Colins, O. F. (2017). Disorganized Amygdala Networks in Conduct-Disordered Juvenile Offenders With Callous-Unemotional Traits. *Biol Psychiatry*, *82*(4), 283-293. doi:10.1016/j.biopsych.2016.05.017
- Aghajani, M., Veer, I. M., van Hoof, M. J., Rombouts, S. A., van der Wee, N. J., & Vermeiren, R. R. (2016). Abnormal functional architecture of amygdala-centered networks in adolescent posttraumatic stress disorder. *Hum Brain Mapp*, *37*(3), 1120-1135. doi:10.1002/hbm.23093
- Baas, D., Aleman, A., & Kahn, R. S. (2004). Lateralization of amygdala activation: a systematic review of functional neuroimaging studies. *Brain Res Brain Res Rev*, *45*(2), 96-103. doi:10.1016/j.brainresrev.2004.02.004
- Eickhoff, S. B., Stephan, K. E., Mohlberg, H., Grefkes, C., Fink, G. R., Amunts, K., & Zilles, K. (2005). A new SPM toolbox for combining probabilistic cytoarchitectonic maps and functional imaging data. *Neuroimage*, *25*(4), 1325-1335. doi:10.1016/j.neuroimage.2004.12.034
- Eickhoff, S. B., Thirion, B., Varoquaux, G., & Bzdok, D. (2015). Connectivity-based parcellation: Critique and implications. *Hum Brain Mapp*, *36*(12), 4771-4792. doi:10.1002/hbm.22933
- Eickhoff, S. B., Yeo, B. T. T., & Genon, S. (2018). Imaging-based parcellations of the human brain. *Nat Rev Neurosci*, *19*(11), 672-686. doi:10.1038/s41583-018-0071-7
- Etkin, A., Prater, K. E., Schatzberg, A. F., Menon, V., & Greicius, M. D. (2009). Disrupted amygdalar subregion functional connectivity and evidence of a compensatory network in generalized anxiety disorder. *Arch Gen Psychiatry*, *66*(12), 1361-1372. doi:10.1001/archgenpsychiatry.2009.104
- Qin, S., Young, C. B., Supekar, K., Uddin, L. Q., & Menon, V. (2012). Immature integration and segregation of emotion-related brain circuitry in young children. *Proc Natl Acad Sci U S A*, *109*(20), 7941-7946. doi:10.1073/pnas.1120408109
- Tang, S., Li, H., Lu, L., Wang, Y., Zhang, L., Hu, X., . . . Huang, X. (2019). Anomalous

functional connectivity of amygdala subregional networks in major depressive disorder. *Depress Anxiety*, 36(8), 712-722. doi:10.1002/da.22901

Zilles, K., & Amunts, K. (2010). Centenary of Brodmann's map--conception and fate. *Nat Rev Neurosci*, 11(2), 139-145. doi:10.1038/nrn2776

REVIEWERS' COMMENTS:

Reviewer #1 (Remarks to the Author):

The authors were very responsive to my comments in their manuscript revision and rebuttal letter. I particularly appreciate the authors effort and rigor in conducting the additional, exploratory analyses I suggested. I realize that this represents a lot of additional work, yet I believe this extra work made the paper more convincing and significant. I read both versions with great interest.

Additional minor comments:

Regarding my past comment #6 in which I asked whether the OCD participants had received prior specialized CBT treatment for OCD: this information should be added to the manuscript in the Methods, Participants section. It could be added at the end of the second paragraph of that section, just after discussing prior medication and wash out period in the OCD participants.

Reviewer #2 (Remarks to the Author):

Authors have properly addressed my points.